# Omics Overview of the SPARC Gene in Mesothelioma

**DOI:** 10.3390/biom13071103

**Published:** 2023-07-11

**Authors:** Licun Wu, Marc de Perrot

**Affiliations:** 1Latner Thoracic Surgery Research Laboratories, Division of Thoracic Surgery, Toronto General Hospital, Toronto General Hospital Research Institute, University Health Network (UHN), 9N-961, 200 Elizabeth Street, Toronto, ON M5G 2C4, Canada; licun.wu@uhnresearch.ca; 2Division of Thoracic Surgery, Princess Margaret Hospital, University Health Network (UHN), Toronto, ON M5G 1L7, Canada; 3Department of Immunology, University of Toronto, Toronto, ON M5S 1A1, Canada

**Keywords:** SPARC, epithelial-mesenchymal transition (EMT), mesothelioma (MESO), omics

## Abstract

The SPARC gene plays multiple roles in extracellular matrix synthesis and cell shaping, associated with tumor cell migration, invasion, and metastasis. The SPARC gene is also involved in the epithelial-mesenchymal transition (EMT) process, which is a critical phenomenon leading to a more aggressive cancer cell phenotype. SPARC gene overexpression has shown to be associated with poor survival in the mesothelioma (MESO) cohort from the TCGA database, indicating that this gene may be a powerful prognostic factor in MESO. Its overexpression is correlated with the immunosuppressive tumor microenvironment. Here, we summarize the omics advances of the SPARC gene, including the summary of SPARC gene expression associated with prognosis in pancancer and MESO, the immunosuppressive microenvironment, and cancer cell stemness. In addition, SPARC might be targeted by microRNAs. Notably, despite the controversial functions on angiogenesis, SPARC may directly or indirectly contribute to tumor angiogenesis in MESO. In conclusion, SPARC is involved in tumor invasion, metastasis, immunosuppression, cancer cell stemness, and tumor angiogenesis, eventually impacting patient survival. Strategies targeting this gene may provide novel therapeutic approaches to the treatment of MESO.

## 1. Introduction

An increasing number of studies of the SPARC (Secreted Protein, Acidic and Rich in Cysteine) gene and protein functions have been published in recent years along with development of multi-omics technology [1,2,3]. It is notable that SPARC may play critical roles in progression and prognosis of malignant pleural mesothelioma (MESO). MESO is known as a rare cancer originating from mesothelial cells that line the surface of thoracic and peritoneal cavities, associated with asbestos exposure [4,5]. MESO was categorized by WHO in 2015 as epithelioid, biphasic, and sarcomatoid for prognostic relevance and treatment decisions. Among the three subtypes of histology, the patients with the sarcomatoid subtype had the worst prognosis compared with those with biphasic or epithelioid subtypes, whereas the patients with the epithelioid subtype had the best prognosis [6]. The efficacy of current therapeutic approaches is limited, and the outcome of this disease is dismal [7]. Therefore, any strategies that may improve therapeutics and prognosis would be of great significance. 

Multi-omics datasets, generated from clinical specimens and shared via different platforms by global researchers, include transcriptomics, genetics/epigenetics, proteomics, metabolomics, and single cell RNA-seq [8,9,10]. The online resources make it possible to look into any particular gene of interest in our research field. Here, we summarize the most recent advances relating to the *SPARC* gene in MESO. 

## 2. Summary of the SPARC Gene

The SPARC gene is located on chromosome 5 of humans, and encodes a secreted protein of the same name (https://www.ncbi.nlm.nih.gov/gene/6678, accessed on 1 May 2022). This protein is a member of the small leucine-rich repeat (SLRP) family of extracellular matrix proteins, which are important for cell–matrix interactions and tissue integrity [11]. The SPARC protein binds to collagen and fibronectin, two extracellular matrix proteins, and plays a role in the formation of the extracellular matrix. It also regulates the assembly of other extracellular matrix proteins, such as laminin. The above functions are essential for maintaining tissue structure and integrity. SPARC is also involved in tissue morphogenesis and development during embryogenesis by regulating cell migration, proliferation, and differentiation in various tissues [12,13]. In addition, SPARC is involved in wound healing and tissue remodeling and can influence cell adhesion, migration, and proliferation [14]. SPARC also regulates inflammation, playing a role in the inflammatory response to injury [15]. The *SPARC* gene is required for the collagen in bone calcification but is also involved in extracellular matrix synthesis and promotion of changes in cell shape [16,17]. This gene product has been associated with tumor metastasis based on changes in cell shape, which can promote tumor cell invasion [18,19]. Three transcript variants encoding different isoforms have been found for the *SPARC* gene (https://www.ncbi.nlm.nih.gov/gene?Db=gene&Cmd=DetailsSearch&Term=6678, accessed on 15 May 2022).

The human protein atlas summary of SPARC is depicted in detail at the human protein atlas website (https://www.proteinatlas.org/ENSG00000113140-SPARC, accessed on 1 May 2022) and the TISIDB web portal for tumor and immune system interaction, which integrates multiple heterogeneous data types (http://cis.hku.hk/TISIDB/index.php, accessed on 1 August 2022). The overall description of the SPARC gene is summarized in Table 1. 

SPARC, which can be selectively expressed by the endothelium in response to certain types of injury, induces rounding in adherent endothelial cells in vitro [20]. Study of the influence of SPARC on endothelial permeability concluded that SPARC regulates endothelial barrier function through F-actin-dependent changes in cell shape, coincident with the appearance of intercellular gaps providing a paracellular pathway for extravasation of macromolecules [21].

SPARC is a protein important to bone calcification [22]. Gene Ontology (GO) annotations related to this gene include calcium ion binding and extracellular matrix binding (Table 2). The general function of SPARC protein appears to regulate cell growth through interactions with the extracellular matrix and cytokines by binding with calcium and copper, several types of collagen, albumin, thrombospondin (*THBS1*), PDGF, and cell membranes. There are two calcium binding sites: an acidic domain that binds 5 to 8 Ca^2+^ with a low affinity and an EF-hand loop that binds a Ca^2+^ ion with a high affinity [23]. 

## 3. SPARC Gene Expression and Its Prognostic Value in Pancancer and Mesothelioma

SPARC gene expression in 32 types of cancer can be found in the TCGA database (http://www.emtome.org/ accessed on 1 May 2022) (Figure 1A). Gene expression of SPARC is associated with overall survival of MESO and pancancer. Survival maps of 32 types of cancer show that SPARC gene is a strong prognostic indicator for MESO (Figure 1B,C). Higher expression of SPARC gene is associated with poorer overall survival (Log rank *p* = 3.6 × 10^−5^) and disease-free survival (Log rank *p* = 0.023) in the MESO cohort (Figure 1D,E) and in pancancer (Figure 1F,G). The prognostic value of the EMT gene signature in MESO has been demonstrated elsewhere and by our group [24].

## 4. Previous Studies of SPARC/Osteonectin on Prognosis in MESO and Other Cancers

Previous studies demonstrated the prognostic significance of the secreted protein SPARC, which could be a prognostic biomarker in MESO in a proteomics-based approach [25]. Kao et al. used a proteomics-based approach to identify secreted protein SPARC as a prognostic biomarker in MESO. The SPARC protein, also known as osteonectin (ON, ONT) or basement-membrane protein 40 (BM-40), is a glycoprotein in the bone that binds calcium and collagen [26]. An integrated gene expression analysis was performed on RNA-seq data of MESO patients from the TCGA dataset and from cell lines, and results showed the *SPARC* gene to be overexpressed in MESO [27]. 

Another study showed that high tumor cell platelet-derived growth factor receptor beta (PDGFRB) and stromal *SPARC* expression remained independently associated with shorter survival in MESO, indicating that PDGFRB and SPARC may be potential markers for risk stratification and as targets for therapy [28]. A meta-analysis demonstrated that SPARC expression has prognostic significance in breast cancer [29]. Methylation-mediated SPARC expression was shown to correlate with tumor progression and poor prognosis of breast cancer. Overexpression of SPARC may promote cancer cell migration and invasion, and thus function as an oncogene and a potential therapeutic target for breast cancer [30]. The relationship between stromal-cell-derived SPARC and its clinicopathologic significance was reported in human gastric cancer tissue [31]. The expression of SPARC together with other genes such as FN1 and SERPINE1 may predict poor prognosis of gastric adenocarcinoma [32]. SPARC expression is also associated with tumor metastasis and poor prognosis in head and neck cancers. Exogenous SPARC and SPARC overexpression enhances the EMT signaling pathway via AKT activation and may be associated with tumor progression in head and neck cancers [33]. SPARC is highly expressed in tumor stroma and tumor-associated fibroblasts in pancreatic cancer, and its overexpression in this compartment is associated with poorer prognosis [34]. 

A systemic review and meta-analysis showed prognostic value of SPARC in hepatocellular carcinoma [35]. In summary, SPARC expression is associated with worse survival in MESO, breast cancer, head and neck cancer, gastric cancer, and hepatocellular carcinoma.

## 5. SPARC and the Immunosuppressive Tumor Microenvironment

Stromal cells and extracellular matrix are the major components contributing to the interaction of tumor cells with their microenvironment. The immunosuppressive microenvironment drives non-malignant cells to change phenotypic plasticity, which promotes tumor cell aggressiveness and invasion [36]. Myeloid-derived suppressor cells (MDSCs) are well-known key negative regulators of the immune response during tumor growth. SPARC was considered a new MDSC marker licensing suppressive activities. SPARC is identified in both human and mouse MDSCs with immune suppressive capacity and pro-tumoral activities, including the induction of EMT and angiogenesis [37]. TCGA data showed a strong positive correlation between gene co-expression of SPARC and MDSC infiltration in MESO (Figure 2A). Considerable evidence has shown that SPARC mediates the TGF-β1 signaling pathway in different cancer types. TCGA data also showed a strong positive correlation between gene co-expression of SPARC and TGF-β1/TGF-βR1 (Figure 2B,C) in MESO and other cancer types (http://cbiolportal.org/ accessed on 1 August 2022). TGF-β1-induced EMT promotes targeted migration of breast cancer cells towards lymphatic vessels [38]. SPARC is a key mediator of TGF-β1-induced renal cancer metastasis [39]. SPARC acts as a mediator of TGF-β1 in promoting EMT in lung cancer [40]. 

The function of SPARC has been shown to act as a mediator of fibrosis [41]. A recent study demonstrated that SPARC in cancer-associated fibroblasts (CAFs) is an independent indicator for poor prognosis in non-metastatic triple-negative breast cancer and exhibits pro-tumor activity, suggesting that patients with SPARC-expressing CAFs could be eligible for anti-SPARC targeted therapy [42]. TCGA data also showed a strong positive correlation between gene co-expression of SPARC and CAF infiltration in MESO (Figure 2D). TGF-β1-activated CAFs promote breast cancer invasion, metastasis, and EMT [43]. EMT is induced by various signaling pathways, including TGF-β1, NOTCH, and receptor tyrosine kinases. EMT is a crucial mechanism governing the current classification of epithelioid, biphasic, and sarcomatoid diffuse MESO. The mesenchymal epithelial transition factor gene (c-MET) encodes a transmembrane protein c-MET receptor tyrosine, which is primarily activated by its ligand, hepatocyte growth factor (HGF), also known as scatter factor. HGF binding to the c-MET receptor leads to the activation of PI3K/AKT, MAPK/ERK, and STAT pathways involved in cell proliferation, motility, and angiogenesis in MESO and other types of cancer. Upregulation of MET and mutation in the *c-MET* Semaphorin (SEMA) domain was observed in more aggressive epithelioid MESO [44]. Moreover, the association of MET and SPARC has been observed in different tumors such as esophageal carcinomas, as described by H. Porte et al. [45]. Using single-cell RNA sequencing to investigate EMT signaling cross-talk and gene regulatory networks, Deshmukh et al. found that the NOTCH signaling pathway acts as a key driver of TGF-β-induced EMT [46].

Dysregulated fatty acid metabolism may contribute to tumorigenesis through interaction with oncogenic signaling. Bellenghi M et al. demonstrated that the stearoyl-CoA desaturating enzyme SCD5 overexpresses in a metastatic clone of 4T1 murine breast cancer cells. Their results showed SCD5-driven reprogramming of fatty acid metabolism was able to block SPARC secretion and eventually reverse the EMT process. More intriguingly, variation in the fatty acid profile by SCD5-gene transduction or the direct administration of oleic acid reduced the immune suppressive activity of MDSCs, eventually enhancing T cell activation. A less immunosuppressive microenvironment generated by SCD5 overexpression was enhanced in SPARC-KO mice, indicating that both extracellular and endogenous SPARC additively regulate MDSCs’ suppressive activities [47].

## 6. SPARC and Cancer Cell Stemness

The SPARC-secreted glycoprotein is produced in various tissues and cells [48]. Studies have shown that SPARC is associated with increased cell motility, invasion, and migration, as well as increased resistance to chemotherapy and radiation [49]. Additionally, SPARC is believed to be involved in the invasion and metastasis of MESO and is known to be upregulated in MESO tumors [31]. Evidence has shown that SPARC promotes self-renewal of limbal epithelial stem cells (LESCs) and ocular surface restoration through JNK and p38-MAPK signaling pathways. Zhou et al. confirmed that the secreted protein SPARC was able to promote the proliferation and suppress the spontaneous differentiation of LESCs in vitro [50]. On the contrary, loss of SPARC could protect hematopoietic stem cells from chemotherapy toxicity by accelerating their return to quiescence [51]. As a matricellular protein SPARC, secreted by a clonal tumor cell subpopulation displaying non-cancer stem cell (CSC) properties in prostate cancer, is a paracrine factor exerted on a distinct tumor cell subpopulation enriched in CSCs. This paracrine interaction enhanced metastatic behavior of the CSC-enriched cancer cell subpopulation. SPARC is expressed in primary prostate cancer cells and metastatic samples, and thus could be a tumor progression biomarker and a therapeutic target in advanced prostate cancer [52]. 

Mesenchymal stem cells induce tumor stroma formation and epithelial-mesenchymal transition (EMT) through SPARC expression in colorectal cancer [53]. Targeting cancer stem cell signature gene SMOC-2 overcomes chemoresistance and inhibits cell proliferation of endometrial carcinoma [54]. However, it remains unknown whether the SPARC gene is a promoter of mesothelioma cell stemness.

## 7. SPARC Identified as a MESO-Specific EMT Gene

The SPARC gene is found to be upregulated in a variety of cancer types, including MESO [55]. The increased expression of the SPARC gene in MESO is associated with increased epithelial-to-mesenchymal transition (EMT), which is a process by which epithelial cells undergo a transformation to become more motile and invasive [56]. This is a critical step in the progression of MESO, as it allows the tumor cells to spread to other parts of the body. The exact mechanism of how SPARC gene expression leads to EMT in MESO is not yet fully understood. Studies have shown that the SPARC protein can interact with other proteins to alter gene expression and cell signaling pathways, which can in turn lead to EMT [57]. In addition, studies have found that the expression of SPARC can be regulated by other factors such as hypoxia, inflammation, and transforming growth factor-beta (TGF-beta) [58]. Overall, the SPARC gene is upregulated in MESO and is involved in EMT, which is essential for the progression of the disease. Further research is needed to understand how SPARC gene expression contributes to EMT in MESO and how it can be used as a therapeutic target.

MESO prognosis is dismal, especially for those with non-epithelioid histology including sarcomatoid and biphasic subtypes. Overexpression of the SPARC gene has been observed more frequently in non-epithelioid than epithelioid subtype MESO [2]. 

The networks in which the SPARC gene is involved were determined by GeneMANIA (Figure 3A). All networks, including physical interactions, co-expression, predicted, co-localization, genetic interactions, pathways, and shared protein domains, were analyzed based on the published literature (https://genemania.org/search/homo-sapiens/sparc accessed on 1 May 2022). Hallmark gene sets of SPARC-centered networks were determined by gene set enrichment analysis (GSEA) (Figure 3B). Top hallmark gene sets include EMT, angiogenesis, myogenesis, and coagulation. We also identified a panel of emerging EMT gene signatures in MESO with prognostic implications and demonstrated that SPARC is a key EMT gene in MESO [2]. 

The matricellular glycoprotein SPARC mediates the interaction between cells and extracellular matrix. It functions as a regulator of matrix organization by modulating cellular behavior. SPARC over expression was observed in stromal tissues and epithelial cells in most cancers. Fibronectin mediates activation of stromal fibroblasts by SPARC in endometrial cancer cells [59]. The infiltration of cancer-associated fibroblasts (CAFs) was positively correlated with EMT gene SPARC expression, thus increasing the risk of poor survival of MESO [2].

## 8. EMT Genes May Be Targeted by Multiple miRNAs

MicroRNAs (miRNAs, or miRs), a class of regulatory endogenous short RNAs, are involved in various biological functions by targeting mRNA of multiple protein-coding genes, thus impacting signaling pathways [60,61]. Evidence has indicated that upregulation of miR-143 in bladder cancer EJ138 cells leads to inhibition of proliferation and migration, while restoration of miR-143 was negatively associated with the expression levels of metastasis, invasion, and EMT-related genes. Therefore, miR-143 may be a potential therapeutic target for bladder cancer [62]. MiR-29c-3p has been reported to inhibit EMT and the proliferation, invasion, and metastasis of cervical cancer cells by targeting SPARC [63]. The miR-592 expression was significantly elevated in colorectal cancer tissues and cell lines and was associated with patient survival. The results showed that miR-592 acts as an oncogene, partly by directly inhibiting SPARC expression [64]. In addition, in leukemia, small RNA-seq analysis reveals a similar miRNA transcriptome in children and young adults as T cell acute lymphoblastic leukemia, and indicates miR-143-3p as a novel candidate tumor suppressor in this leukemia [65]. Low expression of specific miRNAs has been associated with poor survival in MESO [66,67]. However, there are no studies available so far on MESO-specific EMT markers and their related miR genes. In our previous work, we identified a panel of EMT gene signatures in MESO [2]. From the TCGA database, multiple miRNAs were found to be correlated with EMT genes. SPARC gene expression, specifically, has been found to significantly correlate with 97 miRNAs. After looking at the overlaps of miRNA genes among the EMT genes (http://www.emtome.org/, http://mirdb.org/ accessed on 1 May 2022), we observed a negative correlation of both MIR193A and MIR652 genes with five EMT genes (SPARC, COL5A2, ITGAV, ACTA2, and CALD1), and MIR193A may target extra genes TPM2 and TNFRSF12A, while MIR652 may target TIMP3 and SERPINH1, besides the five EMT genes (Figure 4A,B). Underexpressed miR genes in cancers function as tumor suppressor genes and may inhibit cancers by regulating oncogenes that control cell differentiation or apoptosis. The miR expression profiles may become useful biomarkers for cancer diagnostics and therapeutic approaches. Downregulated expression of anti-tumor miR-383 has been reported in variable cancers. MiR-383 acts as a tumor suppressor by directly targeting the 3′-untranslated region (3′-UTR) of the mRNA of some oncogenes to attenuate cancer cell proliferation, invasion, migration, angiogenesis, immunosuppression, EMT, resistance to chemotherapy, and the development of cancer stem cells, while promoting apoptosis [68]. Therefore, the *MIR193A* and *MIR652* genes might be potential tumor suppressors in MESO, and knock-in of these miR genes may result in significant changes in cellular functions, especially the EMT phenotype via downregulating SPARC gene expression. 

## 9. Controversial Function of SPARC on Angiogenesis

Angiogenesis is the process of forming new blood vessels from existing ones [69]. In MESO, angiogenesis is necessary for the growth of the tumors and the spread of the disease [70]. The SPARC gene is involved in the regulation of angiogenesis in MESO by acting as a regulator of the production of vascular endothelial growth factor (VEGF), which is a protein that stimulates the formation of new blood vessels in MESO tumors [71,72,73]. The fundamental function of SPARC in angiogenesis still remains open, as, in most circumstances, SPARC promotes angiogenesis, whereas in some others SPARC may play a controversial anti-angiogenic activity [61,74]. SPARC overexpression in neuroblastoma cells inhibited neo-vascularization in vivo in a mouse dorsal air sac model. The authors observed endothelial cell death induced by SPARC overexpression by co-localization studies with TUNEL assay and an endothelial marker CD31 in a SPARC-overexpressed xenograft tumor model. Their data collectively suggested that SPARC overexpression induces endothelial cell apoptosis and inhibits angiogenesis both in vitro and in vivo. This study supports the notion that SPARC may play an antiangiogenic role [75]. Another study showed that, in addition to blocking angiogenesis, SPARC may inhibit tumor growth by promoting stromal recruitment. The in vitro study demonstrated that SPARC could increase basic fibroblast growth factor-induced fibroblast migration, thus leading to accumulation of fibroblast in tumor stroma [76]. Chlenski et al. demonstrated that SPARC peptides had potent anti-angiogenic and anti-tumorigenic effects in neuroblastoma [77].

On the contrary, a large number of studies support the promotion by SPARC of tumor angiogenesis. Some evidence has indicated that SPARC induces M2 polarization of macrophages to promote proliferation, migration, and angiogenesis of cholangiocarcinoma cells, whereas downregulation of SPARC prevents the M2 polarization of macrophages. Silencing SPARC resulted in inhibition of the M2 macrophage-mediated effects on tumor cell proliferation, migration, and angiogenesis [78]. Most recently, a study showed that SPARC may be a novel regulator of vascular cell function in pulmonary hypertension [79]. SPARC induces inflammatory interferon response in macrophages during aging [80]. As early as 2006, Kzhyshkowska et al. proposed that alternatively activated macrophages coordinated extracellular matrix remodeling, angiogenesis, and tumor progression via stabilin-1 (*STAB1*)-mediated endocytosis of SPARC, and thereby regulate its extracellular concentration [81]. 

In a study to determine whether modulating the expression of extracellular matrix (ECM) modifiers in mesenchymal stem cells and in conventional glioma cell lines might improve tumor invasion and vascularization, the authors observed that SPARC-like 1 (SPARCL1) expression was increased in both mesenchymal stem cells and glioma cell lines by lentiviral transduction. SPARCL1 is a member of the SPARC family of matricellular proteins, which has been implicated in regulating cell migration, proliferation, and differentiation. They may share significant homology [82]. SPARCL1 expression significantly enhanced the microvascular proliferation and tumor neo-angiogenesis of glioblastoma. SPARCL1 overexpression in experimental glioblastoma resulted in increased microvascular density and angiogenesis [83]. 

To sum up, SPARC may directly or indirectly contribute to tumor angiogenesis; therefore, inhibition of SPARC gene expression might be expected to be an anti-angiogenic approach to the treatment of MESO, as EMT and angiogenesis are the top hallmark networks that the SPARC gene is involved with.

## 10. SPARC May Be a Potential Therapeutic Target in Mesothelioma

The SPARC protein has two calcium binding sites. A study showed that treatment of calcium glucoheptonate resulted in increased proliferation and calcium uptake in the MG-63 cells and elevated expression of osteopontin and osteogenic genes such as collagen-1, SPARC, and osteocalcin [84]. 

Small molecules such as calcium citrate and calcium phosphate have been used to target SPARC by inducing overexpression (https://go.drugbank.com/drugs/ accessed on 1 May 2022). Therefore, inhibition of SPARC gene by interrupting the EMT process would be a therapeutic approach to MESO treatment.

As stated above, some miRs may target the SPARC gene, leading to an inhibition of its function. For instance, *MIR193A* and *MIR652* genes’ knock-in may function as a tumor suppressor by inhibiting SPARC function in the EMT phenotype of MESO. Neutralization of SPARC function by monoclonal antibody could be another potential MESO treatment [35]. 

## 11. Concluding Remarks

In conclusion, the *SPARC* gene is involved in multiple processes of tumor development and progression in cancers including MESO. SPARC makes a contribution to the immunosuppressive microenvironment, which drives the phenotypic plasticity of tumor cells to promote their aggressiveness and invasion. A strong positive correlation was observed between gene co-expression of SPARC and TGF-β1/TGF-βR1 in MESO. SPARC acts as a mediator of TGF-β1 in promoting EMT. SPARC has been shown to be a critical EMT marker of MESO, and its overexpression is highly associated with poor prognosis. Therefore, this gene may be an independent prognostic indicator for MESO, as well as a promising therapeutic target.

## Figures and Tables

**Figure 1 biomolecules-13-01103-f001:**
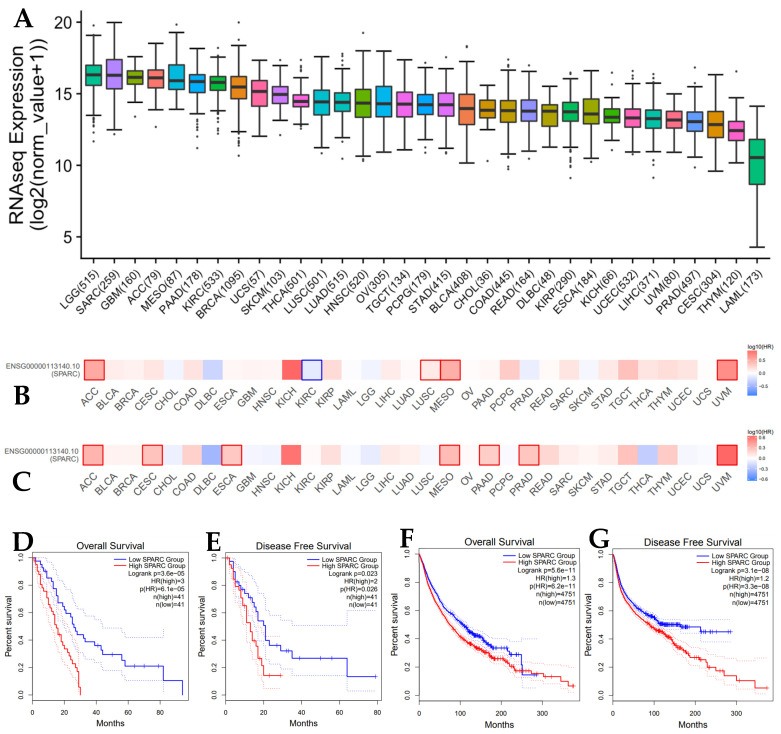
SPARC gene expression and its association with survival in the MESO cohort and pancancer. (**A**) SPARC gene expression in the pancancer TCGA database; (**B**,**C**) survival maps showing that SPARC gene expression is associated with overall survival (OS) and disease-free survival (DFS) in pancancer; (**D**,**E**) SPARC gene expression associated with OS and DFS in the MESO cohort; (**F**,**G**) SPARC gene expression associated with OS and DFS in pancancer.

**Figure 2 biomolecules-13-01103-f002:**
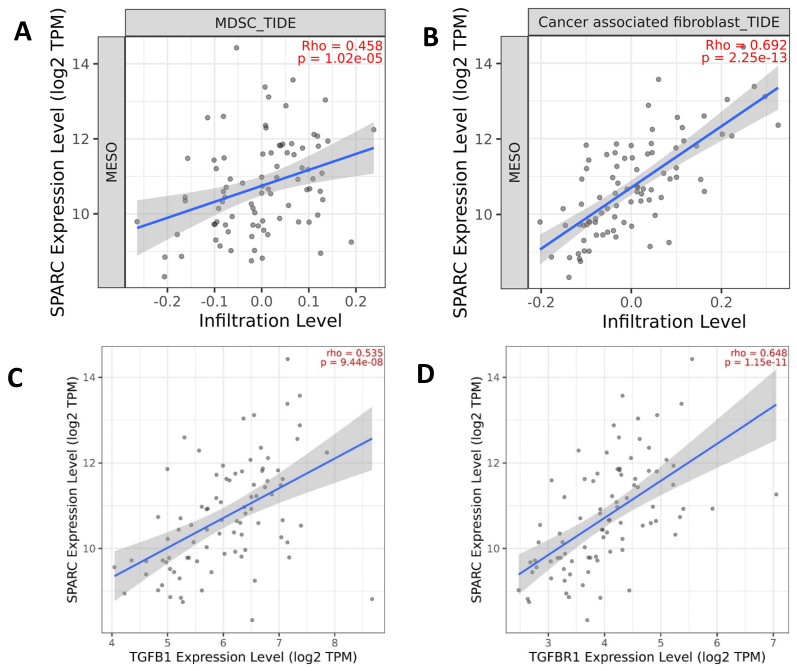
Gene expression of the EMT gene SPARC is correlated with immunosuppressive components. (**A**,**B**) Infiltration levels of myeloid-derived suppressor cells (MDSCs) and cancer-associated fibroblasts (CAFs) are positively correlated with SPARC gene expression; (**C**,**D**) TGFB1/TGFBR1 gene expression is positively correlated with SPARC gene expression. The graphs were generated based on the TCGA database from www.cbiolportal.org accessed on 1 May 2022 and http://timer.cistrome.org/ accessed on 1 May 2022.

**Figure 3 biomolecules-13-01103-f003:**
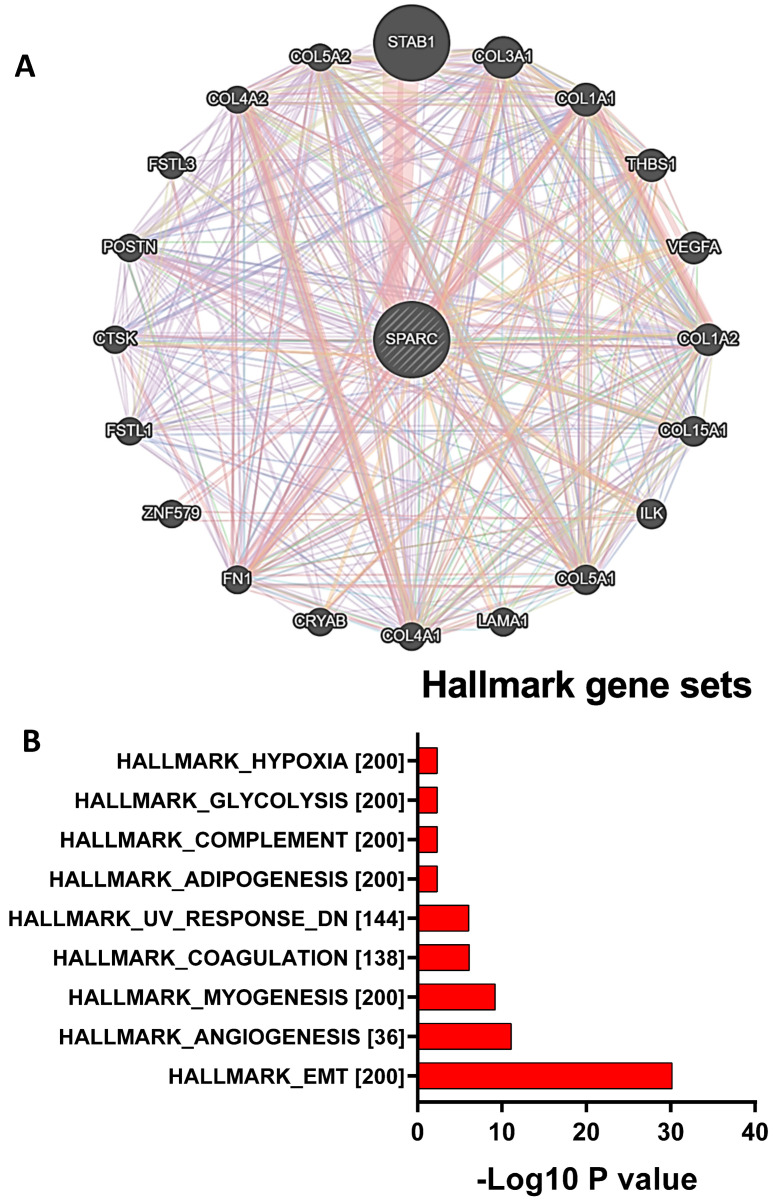
Networks in which the SPARC gene is involved, as determined by GeneMANIA. (**A**) All networks, including physical interactions, co-expression, predicted, co-localization, genetic interactions, pathways, and shared protein domains, are analyzed based on the published literature. https://genemania.org/search/homo-sapiens/sparc accessed on 1 May 2022; (**B**) Hallmark gene sets of SPARC-centered networks determined by gene set enrichment analysis (GSEA).

**Figure 4 biomolecules-13-01103-f004:**
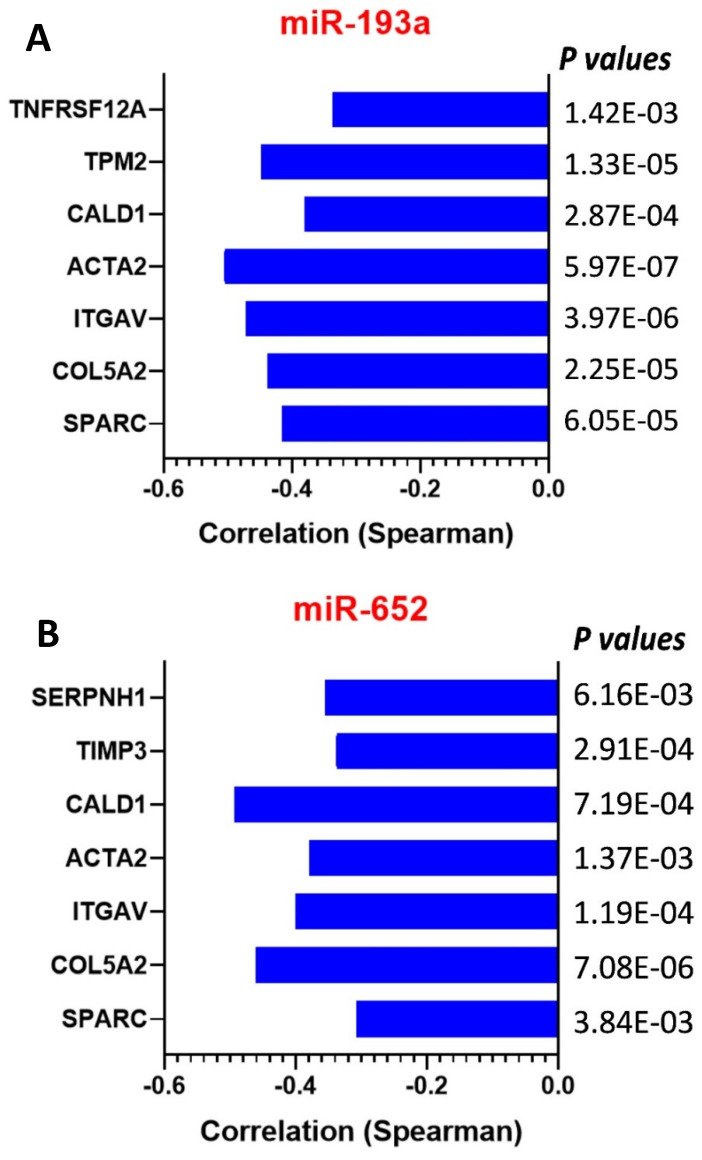
Negative correlation of EMT genes and miRNA gene expression in TCGA MESO. (**A**) The miR-193a (*MIR193A*) gene expression is negatively correlated with 7/9 EMT genes identified in MESO; (**B**) the miR-652 (*MIR652*) gene expression is negatively correlated with 7/9 EMT genes identified in MESO. Five genes may be targeted by both miR-193a and miR-652.

**Table 1 biomolecules-13-01103-t001:** Overview summary of SPARC gene expression.

Human Protein Atlas	Summary of SPARC
**Protein**	Secreted protein acidic and cysteine rich
**Gene name**	SPARC (BM-40, ON, ONT)
**Tissue expression cluster**	Non-specific angiogenesis (mainly)
**Single cell type specificity**	Cell type enhanced (Sertoli cells, smooth muscle cells, endometrial stromal cells, peritubular cells, fibroblasts, theca cells, endothelial cells, adipocytes)
**Single cell type expression cluster**	Fibroblasts—ECM organization (mainly)
**Immune cell specificity**	Group enriched (neutrophil, basophil, classical monocyte)
**Cancer prognostic summary**	Prognostic marker in renal cancer (unfavorable)
**Predicted location**	Intra/extra-cellular, subcellular (vesicles), secreted to blood (different isoforms)
**Ligand (UniProt)**	Calcium, copper, metal-binding
**Chromosomal Location**	5q31-q33
**Domain**	PF09289 Follistatin/Osteonectin-like EGF domain PF00050 Kazal-type serine protease inhibitor domain PF10591 Secreted protein acidic and rich in cysteine Ca binding region
**Protein function (UniProt)**	Appears to regulate cell growth through interactions with the extracellular matrix and cytokines via binding calcium and copper, several types of collagen, albumin, thrombospondin, PDGF, and cell membranes. Two calcium binding sites include an acidic domain that binds 5 to 8 Ca(2+) with a low affinity and an EF-hand loop that binds a Ca(2+) ion with a high affinity.
**Disease involvement**	Cancer-related genes, disease variant, osteogenesis imperfecta
**Gene Ontology**	Top 10 GO annotations BP, MF and CC, See Table 2

**Table 2 biomolecules-13-01103-t002:** Gene ontology top 10 terms that involve the SPARC gene.

Gene Ontology (GO)	GO Terms
Biological process (BP)	GO:0001101 Response to acid chemical
	GO:0001501 Skeletal system development
	GO:0001503 Ossification
	GO:0001525 Angiogenesis
	GO:0001667 Ameboidal-type cell migration
	GO:0001935 Endothelial cell proliferation
	GO:0001936 Regulation of endothelial cell proliferation
	GO:0001937 Negative regulation of endothelial cell proliferation
	GO:0002237 Response to molecule of bacterial origin
	GO:0002576 Platelet degranulation
Molecular function (MF)	GO:0005518 Collagen binding
	GO:0050840 Extracellular matrix binding
Cellular component (CC)	GO:0005578 Proteinaceous extracellular matrix
	GO:0005604 Basement membrane
	GO:0016363 Nuclear matrix
	GO:0030139 Endocytic vesicle
	GO:0030141 Secretory granule
	GO:0030659 Cytoplasmic vesicle membrane
	GO:0030667 Secretory granule membrane
	GO:0031091 Platelet alpha granule
	GO:0031092 Platelet alpha granule membrane
	GO:0031093 Platelet alpha granule lumen

## Data Availability

Not applicable.

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
