# Peer review of "Omics Overview of the SPARC Gene in Mesothelioma"

_biomolecules, 2023, doi:10.3390/biom13071103_

Round 1
Reviewer 1 Report
Omics overview of SPARC gene in mesothelioma
In this review, the authors focus on the role of the SPARC gene/protein in diffuse malignant mesothelioma (DM), taking advantage of the 'multi-omic' approaches used in DM and other cancers in which SPARC has been studied. Despite the multi-omics approach, the topic has not been comprehensively covered. For this reason, this reviewer feels that some points should be better clarified, where possible.
Major revision:
Line 167: Authors statement “EMT is induced by various signaling pathways, including TGF-β1, NOTCH, and receptor tyrosine kinases” is rather generic.
EMT is a crucial mechanism governing the current classification of epithelioid, biphasic and sarcomatoid DMs. This reviewer’s opinion is that this aspect should be expanded. One of the main tyrosine kinase receptor involved in DMs such as Mesenchymal Epithelial Transition gene (c-MET) should be accounted by authors. Also citing a study conducted on peritoneal DMs in which up-regulation of MET was observed as well as mutation in c-MET SEMA domain in more aggressive epithelioid DMs called “progressed” [1]. Moreover, the association of MET and SPARC has been observed in different tumors such as esophageal carcinomas as described by H. Porte and collaborators [2]
1- Belfiore A, Busico A, Bozzi F, et al. Molecular Signatures for Combined Targeted Treatments in Diffuse Malignant Peritoneal Mesothelioma. Int J Mol Sci. 2019;20(22):5817. Published 2019 Nov 19. doi:10.3390/ijms20225817
2- Porte H, Triboulet JP, Kotelevets L, et al. Overexpression of stromelysin-3, BM-40/SPARC, and MET genes in human esophageal carcinoma: implications for prognosis. Clin Cancer Res. 1998;4(6):1375-1382.
Minor revisions:
Line 31,32: The authors refer to malignant pleural mesothelioma as MESO. However, the text speaks of both pleural and peritoneal mesothelioma, which despite the location are neoplasm with differences in frequency, incidence following asbestos exposure and molecular signatures. Please clarify if the review is focused only on pleural DMs, if not the text needs to be remodelled taking these differences into account.
Line 112-114: please uniform the acronym “MPM” with the one chosen for the study “MESO”.
Figure 2 A-D:
Please change order of correlation plots of MDSC_TIDE (A) and CAF_TIDE(D) à MDSC_TIDE (A) and CAF_TIDE (B). Do the same for TGFB1_expression (B) and TGFBR1_expression (C) à TGFB1_expression (C), TGFBR1_expression D. Finally uniform font size of text present in the figure.
Line 237: Figure 3 legend. Add a space between “A)” and “All networks”.
Figure 1, 2, 3:
Please modify the arrangement of the letters identifying the figures, moving them to the top left-hand corner.
Figure 4:
Please add letters for the two correlation plots, they are present in the legend but not in the figure. Add them on the top left-hand corner

Moderate English editing is necessary
Author Response
Dear editors,
Thank you very much for your invitation to contribute this review to the Biomolecules journal. We also appreciate the constructive and valuable comments on our manuscript. We have made changes point-by-point based on the comments made from the reviewers. Please let us know if any further revision is required.
Best regards,
Licun Wu and Marc de Perrot
Reviewer #1
Comments and Suggestions for Authors
Omics overview of SPARC gene in mesothelioma
In this review, the authors focus on the role of the SPARC gene/protein in diffuse malignant mesothelioma (DM), taking advantage of the 'multi-omic' approaches used in DM and other cancers in which SPARC has been studied. Despite the multi-omics approach, the topic has not been comprehensively covered. For this reason, this reviewer feels that some points should be better clarified, where possible.
Major revision:
Line 167: Authors statement “EMT is induced by various signaling pathways, including TGF-β1, NOTCH, and receptor tyrosine kinases” is rather generic.
EMT is a crucial mechanism governing the current classification of epithelioid, biphasic and sarcomatoid DMs. This reviewer’s opinion is that this aspect should be expanded. One of the main tyrosine kinase receptor involved in DMs such as Mesenchymal Epithelial Transition gene (c-MET) should be accounted by authors. Also citing a study conducted on peritoneal DMs in which up-regulation of MET was observed as well as mutation in c-MET SEMA domain in more aggressive epithelioid DMs called “progressed” [1]. Moreover, the association of MET and SPARC has been observed in different tumors such as esophageal carcinomas as described by H. Porte and collaborators [2]
1- Belfiore A, Busico A, Bozzi F, et al. Molecular Signatures for Combined Targeted Treatments in Diffuse Malignant Peritoneal Mesothelioma. Int J Mol Sci. 2019;20(22):5817. Published 2019 Nov 19. doi:10.3390/ijms20225817
2- Porte H, Triboulet JP, Kotelevets L, et al. Overexpression of stromelysin-3, BM-40/SPARC, and MET genes in human esophageal carcinoma: implications for prognosis. Clin Cancer Res. 1998;4(6):1375-1382.
-This is a great point we should address. Some statements have been added to the discussion. The references you pointed out are cited as well.
On page 7, “EMT is a crucial mechanism governing the current classification of epithelioid, biphasic and sarcomatoid diffuse MESO.The mesenchymal epithelial transition factor gene (c-MET) encodes a transmembrane protein c-MET receptor tyrosine, which is primarily activated by its ligand, hepatocyte growth factor (HGF), also known as scatter factor. HGF binding to the c-MET receptor leads to the activation of PI3K/AKT, MAPK/ERK, and STAT pathways involving in cell proliferation, motility, and angiogenesis in MESO and other types of cancer. Up-regulation of MET and mutation in c-MET Semaphorin (SEMA) domain was observed in more aggressive epithelioid MESO (44). Moreover, the association of MET and SPARC has been observed in different tumors such as esophageal carcinomas as described by H. Porte, et al (45).”
Minor revisions:
Line 31, 32: The authors refer to malignant pleural mesothelioma as MESO. However, the text speaks of both pleural and peritoneal mesothelioma, which despite the location are neoplasm with differences in frequency, incidence following asbestos exposure and molecular signatures. Please clarify if the review is focused only on pleural DMs, if not the text needs to be remodelled taking these differences into account.
Line 112-114: please uniform the acronym “MPM” with the one chosen for the study “MESO”.
-Yes, we changed into uniform term- MESO.
Figure 2 A-D:
Please change order of correlation plots of MDSC_TIDE (A) and CAF_TIDE(D) à MDSC_TIDE (A) and CAF_TIDE (B). Do the same for TGFB1_expression (B) and TGFBR1_expression (C) à TGFB1_expression (C), TGFBR1_expression D. Finally uniform font size of text present in the figure.
-The order of the plots has been changed as the reviewer suggested.
Line 237: Figure 3 legend. Add a space between “A)” and “All networks”.
-Corrected.
Figure 1, 2, 3:
Please modify the arrangement of the letters identifying the figures, moving them to the top left-hand corner.
-Changed.
Figure 4:
Please add letters for the two correlation plots, they are present in the legend but not in the figure. Add them on the top left-hand corner
-Letters A and B were added to Fig. 4.

Reviewer 2 Report
The SPARC gene plays multiple roles in cancer, including tumor cell migration, invasion, and metastasis, as well as involvement in the epithelial-mesenchymal transition (EMT) process. Overexpression of SPARC is associated with poor survival in mesothelioma (MESO) patients, suggesting its potential as a prognostic factor in MESO. SPARC's overexpression is also correlated with an immunosuppressive tumor microenvironment. Strategies targeting the SPARC gene may offer novel therapeutic approaches for MESO treatment.
However, the current version of the manuscript has several shortcomings that need to be addressed in order to improve its chances of being accepted for publication.
1. what is the role of SPARC gene in normal tissue (non tumor tissue)?
2. Please, present the survival maps showing the overall survival (OS) and disease-free survival (DFS) in several cancers
The SPARC gene plays multiple roles in cancer, including tumor cell migration, invasion, and metastasis, as well as involvement in the epithelial-mesenchymal transition (EMT) process. Overexpression of SPARC is associated with poor survival in mesothelioma (MESO) patients, suggesting its potential as a prognostic factor in MESO. SPARC's overexpression is also correlated with an immunosuppressive tumor microenvironment. Strategies targeting the SPARC gene may offer novel therapeutic approaches for MESO treatment.
However, the current version of the manuscript has several shortcomings that need to be addressed in order to improve its chances of being accepted for publication.
1. what is the role of SPARC gene in normal tissue (non tumor tissue)?
2. Please, present the survival maps showing the overall survival (OS) and disease-free survival (DFS) in several cancers
Author Response
Reviewer #2
Comments and Suggestions for Authors
The SPARC gene plays multiple roles in cancer, including tumor cell migration, invasion, and metastasis, as well as involvement in the epithelial-mesenchymal transition (EMT) process. Overexpression of SPARC is associated with poor survival in mesothelioma (MESO) patients, suggesting its potential as a prognostic factor in MESO. SPARC's overexpression is also correlated with an immunosuppressive tumor microenvironment. Strategies targeting the SPARC gene may offer novel therapeutic approaches for MESO treatment.
However, the current version of the manuscript has several shortcomings that need to be addressed in order to improve its chances of being accepted for publication.
Thank you very much for your valuable suggestions. Changes have been made accordingly.
- what is the role of SPARC gene in normal tissue (non tumor tissue)?
-The roles of SPARC gene in normal tissues were added to the text. Please see pages 3-4 including 2 references:
“The roles of SPARC gene in normal tissues were included in the text on page 3-4. The SPARC protein binds to collagen and fibronectin, two extracellular matrix proteins, and plays a role in the formation of the extracellular matrix. It also regulates the assembly of other extracellular matrix proteins, such as laminin. The above functions are essential for maintaining tissue structure and integrity. SPARC is also involved in tissue morphogenesis and development during embryogenesis by regulating cell migration, proliferation, and differentiation in various tissues (12, 13). In addition, SPARC is involved in wound healing and tissue remodeling and can influence cell adhesion, migration and proliferation (14).”
12 Barker TH, Baneyx G, Cardó-Vila M, Workman GA, Weaver M, Menon PM, Dedhar S, Rempel SA, Arap W, Pasqualini R, Vogel V, Sage EH. SPARC regulates extracellular matrix organization through its modulation of integrin-linked kinase activity. J Biol Chem. 2005 Oct 28;280(43):36483-93. doi: 10.1074/jbc.M504663200. Epub 2005 Aug 22. PMID: 16115889.
13 Lane TF, Sage EH. The biology of SPARC, a protein that modulates cell-matrix interactions. FASEB J. 1994 Feb;8(2):163-73. PMID: 8119487.
- Please, present the survival maps showing the overall survival (OS) and disease-free survival (DFS) in several cancers
-New Figure 1 includes SPARC gene expression associated with OS and DFS curves (F&G) in pancancer (n=4751 of SPARC high expression and n=4751 low expression) from TCGA data.

Round 2
Reviewer 1 Report
The authors have modified the text in line with this reviewer's request.